# Molecular Detection of Antimalarial Drug Resistance in *Plasmodium vivax* from Returned Travellers to NSW, Australia during 2008–2018

**DOI:** 10.3390/pathogens9020101

**Published:** 2020-02-05

**Authors:** Chaturong Noisang, Wieland Meyer, Nongyao Sawangjaroen, John Ellis, Rogan Lee

**Affiliations:** 1Marie Bashir Institute for Infectious Diseases and Biosecurity, The University of Sydney, Sydney, NSW 2006, Australia; chaturongmt@gmail.com (C.N.); wieland.meyer@sydney.edu.au (W.M.); 2Westmead Institute for Medical Research, Westmead, NSW 2145, Australia; 3Westmead Hospital (Research and Education Network), Westmead, NSW 2145, Australia; 4Department of Microbiology, Faculty of Science, Prince of Songkla University, Hat Yai, Songkhla 90110, Thailand; nongyao.s@psu.ac.th; 5School of Life Sciences, University of Technology Sydney, Ultimo, NSW 2007, Australia; John.Ellis@uts.edu.au; 6Centre for Infectious Diseases and Microbiology Laboratory Services, NSW Health Pathology-ICPMR, Westmead Hospital, Westmead, NSW 2145, Australia

**Keywords:** *plasmodium vivax*, drug resistance, chloroquine, sulfadoxine-pyrimethamine

## Abstract

To monitor drug resistance in *Plasmodium vivax*, a multidrug resistance 1 (*Pvmdr1*) gene and a putative transporter protein (*Pvcrt-o*) gene were used as molecular markers for chloroquine resistance. The biomarkers, the dihydrofolate reductase (*Pvdhfr*) gene and the dihydropteroate synthetase (*Pvdhps*) gene, were also used for the detection of resistance to sulphadoxine-pyrimethamine (SP); this drug is often accidentally used to treat *P. vivax* infections. Clinical blood samples (n **=** 120) were collected from patients who had been to one of eight malaria-endemic countries and diagnosed with *P. vivax* infection. The chloroquine resistance marker, the *Pvmdr1* gene, showed **F**976:**L**1076 mutations and **L**1076 mutation. A **K**10 insertion in the *Pvcrt-o* gene was also found among the samples successfully sequenced. A combination of **L/I**57:**R**58:**M**61:**T**117 mutations in the *Pvdhfr* gene and **G**383:**G**553 mutations in the *Pvdhps* gene were also observed. Mutations found in these genes indicate that drug resistance is present in these eight countries. Whether or not countries are using chloroquine to treat *P. vivax*, there appears to be an increase in mutation numbers in resistance gene markers. The detected changes in mutation rates of these genes do suggest that there is still a trend towards increasing *P. vivax* resistance to chloroquine. The presence of the mutations associated with SP resistance indicates that *P. vivax* has had exposure to SP and this may be a consequence of either misdiagnosis or coinfections with *P. falciparum* in the past.

## 1. Introduction

In Australia, about 500 malaria cases are reported each year [1] and just under half of them are diagnosed with *Plasmodium vivax* [2]. All cases are in travellers who have been to malaria-endemic countries. *Plasmodium falciparum* remains the predominant malaria species, but *P. vivax* has become the major cause of human malaria in endemic regions outside of Africa, such as Asia, Central and South America and Oceania. Globally, it is estimated that there are between 80 million and up to 400 million vivax malaria cases each year [3,4,5]. Although malaria control and elimination programs are promoted in these countries, these programs are being hindered by the emergence of drug-resistant strains of *P. vivax*.

Usually, chloroquine-sensitive strains of *P vivax* are cleared from the blood within 48 h after a patient receives a standard dose of 25 mg/kg chloroquine [6]. However, chloroquine-resistant strains can recrudesce within 14 days after the start of treatment. A recurrent parasitemia with 100 ng/mL of the measured chloroquine concentration in whole blood on or before the 28th day of treatment is classified as resistant [3,6]. Chloroquine resistance in *P. vivax* was found around the late 1980s and resistance was first reported in Australian travellers, who were diagnosed with *P. vivax* infection after returning from Papua New Guinea (PNG) [7,8]. These travellers failed to clear the parasitemia by the standard chloroquine dose [3,6,9,10]. During this period, other reports of chloroquine-resistant *P. vivax* arose from Asian and South American countries [9].

For *P. falciparum*, the chloroquine-resistant phenotype is strongly associated with mutations in a *P. falciparum* chloroquine resistance transporter (*Pfcrt*) gene and a *P. falciparum* multidrug-resistant transporter-1 (*Pfmdr1*) gene. Studies demonstrated that the *Pfcrt* gene encodes a transmembrane protein located on the parasite digestive vacuole [11,12]. A strain of *P. falciparum* with a mutation of the gene *Pfcrt* alters the movement of CQ away from heme into the parasite digestive vacuole. As a result, the parasite survives under therapeutic CQ levels [12,13]. Unlike *P. falciparum*, chloroquine-resistant *P. vivax* has been extremely difficult to investigate due to the characteristics of this parasite, which include the presence of dormant forms in the liver, some cases with very low parasitaemia, no effective long-term in vitro assay, and the inability to distinguish between relapse, recrudescence and reinfection in therapeutic efficacy studies [14,15]. Therefore, the mechanisms of CQ resistance in *P. vivax* remain unclear. However, studies found correlations between chloroquine-resistant *P. vivax* phenotype and mutations in the multidrug resistance 1 (*Pvmdr1*) gene, and the putative transporter protein (*Pvcrt-o*) gene. These genes are orthologous to *Pfmdr1* and *Pfcrt* genes in *P. falciparum* [3,16,17,18]. The *Pvmdr1* gene is located on chromosome 10, containing 24 single nucleotide polymorphisms (SNPs), while the *Pvcrt-o* gene is located on chromosome 1, contains 5 SNPs and a lysine insertion at the amino acid position 10 [19]. Although molecular methods may not be the most effective tool to monitor drug resistance in *P. vivax*, well-characterised molecular biomarkers do allow for retrospective studies to be done so that molecular changes can be monitored. The multidrug resistance 1 (*Pvmdr1*) gene and the putative transporter protein (*Pvcrt-o*) gene are the recommended markers to monitor chloroquine resistance in *P. vivax*. 

Sulfadoxine-pyrimethamine (SP) combined with other drugs was widely used in the past [20] and remains in use in a few countries such as Iran, Saudi Arabia, Yemen and India to treat patients with uncomplicated *P. falciparum* infections [21]. However, *P. vivax* can be exposed to SP either as a coinfection with *P. falciparum* or following a misdiagnosis of malaria species. Furthermore, SP was suggested as an alternative treatment for vivax malaria in some areas [22], specifically those locations with high rates of mixed infections [23]. Reports have shown that genetic-resistant strains of *P. vivax* to SP treatment can be identified by the detection of mutations present in the dihydrofolate reductase (*Pvdhfr*) and dihydropteroate synthetase (*Pvdhps*) genes [24]. Pyrimethamine resistance correlates with specific single nucleotide polymorphisms of the *Pvdhfr* gene, which results in decreased enzyme affinity to pyrimethamine [24,25], whereas sulfadoxine resistance is associated with nucleotide polymorphisms within the *Pvdhps* gene [24]. This study aimed to detect the prevalence of molecular resistance markers for chloroquine, sulfadoxine and pyrimethamine in *P. vivax*-infected travellers returning to NSW, Australia.

## 2. Results

All *P. vivax* isolates came from infected travellers returning to Australia from India, PNG, Pakistan, Indonesia, Thailand, Solomon Islands, South Korea and Cambodia. Analyses for the presence of resistance biomarkers were undertaken based on geographical region, as numbers from individual countries were low and some would be poorly represented in this survey (see Appendix A). The majority of our patients were male (89/120) and most were in the 21–30 years age group (27/88) (see Table 1). Only 23 females were present in this study with most in the 21–30 and over 61 years age groups. Nine samples had no gender recorded.

### 2.1. Sequence Analysis of The Multidrug Resistance 1 (Pvmdr1) Gene, and The Putative Transporter Protein (Pvcrt-o) Gene

*Pvmdr1* amplicons (n **=** 109) were successfully sequenced and displayed mutations at codon 976 and 1076. As shown in Table 2, double **F**976:**L**1076 mutations were observed in all regions: Oceania (23/37), Southeast Asia (5/10) and South Asia (2/73). In Oceania, PNG showed the highest prevalence of mutations, with 22/29 of isolates (see Appendix A). The single **L**1076 mutation was also detected in all three regions: South Asia (50/73), Oceania (8/37) and Southeast Asia (5/10). This mutation showed the highest prevalence in India (36/51) and Pakistan (14/17) (see Appendix A). All single **F**976 mutations were found in India (7/51) (see Appendix A) and nine isolates from the South Asia region were identified as wildtype. 

A **K**10 insertion in the *Pvcrt-o* gene was identified in nine isolates from India (South Asia) and two isolates from Southeast Asia. No other isolate from this study group had this insertion (see Table 2).

### 2.2. Sequence Analysis of The Dihydrofolate Reductase (Pvdhfr) Gene

A total of 112 sequences were amplified using primers targeting the *Pvdhfr* gene. The majority of the isolates expressing the wildtype genotype were from South Asia (43/53), while a few were present in Oceania (7/37) and Southeast Asia (3/10) (see Table 3). Mutations were observed at codons 57, 58, 61, 117 and 173. A combination of **L**/**I**57:**R**58:**M**61:**T**117 mutations was found in Oceania (21/37), Southeast Asia (4/10) and South Asia (2/73) (see Table 3). In Oceania, many isolates (19/33) from PNG carried these mutations (see Appendix A). Two mutations at **R**58:**N**117 were detected in the three regions South Asia (18/73), Southeast Asia (2/10) and Oceania (1/37). Amongst the South Asian isolates, India (13/50) and Pakistan (5/17) had similar proportions of the **R**58:**N**117 mutations (see Appendix A). The other two mutations at **L**57:**R**58 were observed in Oceania (5/37) and South Asia (1/73) in low numbers and the single mutation at **N**117 was also found in a few samples from South Asia (3/73), Oceania (1/37) and Southeast Asia (1/10).

### 2.3. Sequence Analysis of The Dihydropteroate Synthetase (Pvdhps) Gene

Amplicons (n **=** 116) that were successfully sequenced had mutations at codons 382, 383, 512, 553 and 585. The wildtype genotype was found in all three regions, the majority of which were in South Asia (62/73), Oceania (35/37) and small numbers in Southeast Asia (5/10). Two mutations at **G**383:**G**553 loci were found in South Asia (6/73) (all six came from India), Southeast Asia (2/10) and Oceania (1/37). Isolates showing the single **G**383 mutation were identified in South Asia (2/73) and Southeast Asia (1/10). The single **G**553 mutation was only found in one isolate from South Asia and one from Southeast Asia.

### 2.4. The Prevalence of Tandem Repeat Variants in the Dihydrofolate Reductase (Pvdhfr) Gene

Tandem repeat variants found in the *Pvdhfr* gene locus have been described by different laboratories [26]; the typing of these variants in this study followed that of a recent Thai study from 2018 [7]**.** This group described three different tandem repeat variants found in the *Pvdhfr* gene sequence. The reference strain (Accession No. X98123.1) [27] contains three repeat sets of four amino acids (5’-GGDN-3’) between codons 88 and 103 and was designated as Type 1 or wildtype. The tandem repeat sequence that lacks the six amino acid at codons 98–103 was designated as Type 2. Type 3 sequences have an additional six amino acids inserted between codons 103 and 104. 

Type 1 (wildtype) was the predominant sequence found in all regions (see Table 4). Isolates showing Type 1 sequences were from India (44/51), PNG (16/33), Pakistan (11/17) and Indonesia (4/5). A few isolates had Type 2 repeat variants found in South Asia (9/68) and Southeast Asia (3/10). Type 3 tandem repeats were found mainly in isolates from PNG (17/33) and a few isolates from South Asia (4/68) and Southeast Asia (1/10).

## 3. Discussion

About 30 years ago, the first chloroquine resistance in *P. vivax* was found [8] and subsequently, it appeared in several other countries such as the Solomon Islands, Indonesia, Malaysia, Thailand, Ethiopia, Brazil and Peru [15]. Molecular markers of chloroquine resistance in *P. vivax*, *Pvmdr1* and *Pvcrt-o* genes were used as they are orthologous to *Pfmdr1* and *Pfcrt* genes, which are associated with chloroquine resistance in *P. falciparum* [6]. Among the studied samples herein, the presence of the **F**976 and **L**1076 mutations in the *Pvmdr1* gene and the **K**10 insertion in the *Pvcrt-o* gene were shown previously to be related to chloroquine resistance in *P. vivax* by in vivo and in vitro studies [5,7,16,28,29]. 

This study obtained sequences from both *Pvmdr1* (109/120) and *Pvcrt-o* (88/120) genes and mutations were found in all geographical regions of South Asia, Southeast Asia and Oceania. The **F**976:**L**1076 mutations were more frequent in Oceania (23/37) in comparison to the Southeast Asian (5/10) and South Asian (2/73) regions. A large number of isolates (50/73) from South Asia carried only the **L**1076 mutation and a few isolates from the Southeast Asia (5/10) and Oceania (8/37) regions. The single **F**976 mutation was detected only in seven isolates from the South Asian region. 

The finding of the **F**976:**L**1076 mutations in Oceania is, however, different from earlier studies in PNG between 2006 and 2011; these studies showed the presence of the **F**976 mutation only [30,31,32]. This suggests that there was a rise in the number of chloroquine resistance markers since the first reported case of resistant *P. vivax* in the region, even though chloroquine has not been recommended in the antimalarial drug policy since 2009 [33]. In addition, the high prevalence of the **L**1076 mutation in the South Asian region was consistent with other recent studies in India and Pakistan with respective mutations of 54.5% and 98.0% [34,35]. Only two isolates carried the **F**976:**L**1076 mutations in the Indian study [34], whereas the previous study in Pakistan did not find either the **F**976 mutation or the **F**976:**L**1076 mutations [35]. Furthermore, if the hypothesis of a two-step mutation trajectory is true, then the **L**1076 mutation occurs first, followed by the **F**976 mutation before the emergence of chloroquine resistance [17,36,37]. The finding in the present study with the double **F**976:**L**1076 mutations is an indicator that fully developed chloroquine-resistant strains of *P. vivax* are present. A case report of a chloroquine-resistant *P. vivax* infection in a pregnant woman living in Pakistan [38] would further support the molecular finding in this study although mutations in the *Pvmdr1* gene were not investigated. The increased rate of these mutations suggests that chloroquine resistance in *P. vivax* will become more common in the South Asia region where chloroquine has remained the first-line drug [39]. Our findings from clinical isolates from Southeast Asia were also similar to other studies in this location [5,6,7,16,40]. A clinical report of chloroquine-resistant *P. vivax* in a Thai pregnant woman indicated that recurrent parasites during Day 21 and 143 carried the double **F**976:**L**1076 mutation [40]. This double mutation was found in Thailand in 2008 (49.2%) and again in 2014 (18.4%) [7]. The same double mutations were found in 15.4% of samples in Myanmar between 2009 and 2016 [5]. This could indicate that chloroquine-resistant *P. vivax* is not only present in Thailand but also in neighbouring Myanmar. 

Unlike other well-described mutations in the *Pvmdr1* gene, the role of the **K**10 insertion in *Pvcrt-o* in chloroquine resistance in *P. vivax* is unclear because this mutation is present in some studies and absent in others [3,5,7,16,34,41]. In Thailand, the **K**10 insertion was found in 2008 [16], but was not detected again until recently [7,42]. In our study, Indian samples consistently showed the insertion from 2013 to 2017 (see Appendix A). In addition to variations in finding the **K**10 insertion in the *Pvcrt-o* gene, a recent study suggests that the absence of this mutation (wildtype sequence) in isolates is associated with chloroquine resistance [43]. This would indicate that in our study, the 77 isolates from all three regions which carry the wildtype *Pvcrt-o* gene are of the chloroquine-resistant genotype.

The findings herein show that the chloroquine-resistant-associated genotypes are present in South Asia, Southeast Asia and Oceania, but chloroquine remains the drug of choice for treatment against *P. vivax* in India, Pakistan, Thailand and South Korea [21]. Reports of resistant genotypes [38,40,44,45,46] and the increased mutation frequency found in this study do suggest that resistant phenotypes will continue to emerge in these regions.

A combination of sulfadoxine and pyrimethamine (SP) with other antimalarial drugs is not recommended for treating *P. vivax* [46]. Yet this species of malaria shows emerging mutant genotypes suggesting exposure to SP drug pressures [5,7]. In order to identify SP resistance in *P. vivax*, mutations in the *Pvdhfr* and *Pvdhps* genes were used. These genes are orthologues to *Pfdhfr* and *Pfdhps* genes in SP-resistant *P. falciparum* [5,7,10]. Furthermore, these gene targets have been correlated with SP resistance in *P. vivax* [47]. An in vitro study using yeast models which contained the *Pvdhfr* gene insertion showed that the IC_50_ value of that yeast expressing the **L**/**I**57:**R**58 mutations was sevenfold higher than yeast expressing the wildtype gene [48]. Furthermore, genetically modified yeasts containing **R**58:**N**117 mutations and those with triple mutations of the *Pvdhfr* gene were 200–300-fold more resistant to SP than those with the wildtype gene [48]. The **L**/**I**57:**R**58:**M**61:**T**117 mutations in the *Pvdhfr* gene showed a high level of pyrimethamine resistance as the IC_50_ value was over 500-fold in this yeast model [48]. This quadruple mutation was also linked to higher rates of therapeutic failure of SP in a volunteer group of patients than those isolates with wildtype, single, double and triple mutations in their gene sequences [49].

The present study found the **L**/**I**57:**R**58:**M**61:**T**117 mutations in a large number of isolates from Oceania (21/37), which was followed by the Southeast Asian (4/10) and South Asian (2/73) regions. Our findings were consistent with other studies from these regions [5,7,32,50,51,52]. An increase in treatment failure by 12% occurred in *P. vivax* cases in PNG when the malaria treatment policy in 2000 changed to recommending the use of SP with either chloroquine or amodiaquine as the first-line drug for uncomplicated malaria [53]. India, the largest country in the South Asian region, continues to use SP with other drugs to treat uncomplicated *P. falciparum* malaria [23]. In the South Asian region, the **R**58:**N**117 mutations were present (18/73) in similar numbers found in past studies confirming the higher prevalence of the **R**58:**N**117 mutations (28.16%) than the **L**/**I**57:**R**58:**M**61:**T**117 mutations (19.71%) [51]. It was hypothesised that the **N**117 mutation occurs in parasites as an early development towards SP resistance [54]. This is followed by the **R**58 mutation, while the other two mutations might occur independently [55]. Therefore, a rise in early stages of SP-resistant mutations indicates ongoing SP drug pressure on *P. vivax* in the South Asian region.

Tandem repeat variants were found during the analyses of the *Pvdhfr* gene in the present study. The predominance of the tandem repeat Type 1 (wildtype sequence) occurs in three regions tested and the presence of low numbers of Type 2 was found in South Asia (9) and Southeast Asia (3). Type 3 variants, which contain four repeat sets of 5’-GGDN-3’, were identified in Oceania (19), South Asia (4) and Southeast Asia (1). Other studies have found that all three types are present in India [54,55], whereas Indonesia, Thailand and South Korea have only Type 1 and Type 2 [7,52]. The role of the tandem repeat variants in SP resistance remains unclear [7,56] as these variants are not present in the *Pfdhfr* gene in *P. falciparum* [26,57]. However, Type 3 variants were exclusively linked to the **L**/**I**57:**R**58:**M**61:**T**117 mutations, which are associated with pyrimethamine resistance in *P. vivax*. There have been suggestions that these variants are involved in the development of mutations and higher levels of the resistance and therefore may be useful molecular markers [20,51]. Interestingly, 22/24 isolates carrying the Type 3 variant in this study had the **L**/**I**57:**R**58:**M**61:**T**117 mutations (see Appendix A).

A large number of the studied isolates herein (103/120) from all three regions were found to be the wildtype *Pvdhps* genotype. However, nine isolates from South Asia (6/73), Southeast Asia (2/10) and Oceania (1/37) had the **G**383:**G**55 double mutations, which are linked to sulfadoxine resistance in *P. vivax* [5,7,58]. The current results correlate with the findings made in previous studies from these regions [7,31,32,51,52,55,59]. Some studies [47,60] showed that the mutations in the *Pvdhps* gene were more than twice as likely to emerge in isolates with multiple mutations in the *Pvdhfr* gene. However, only 3/120 of the isolates in the current study were found to carry the **G**383:**G**55 mutations when multiple mutations were present in the *Pvdhfr* gene (see Appendix A).

Gene mutations will be influenced by the different drug policies of each country and these decisions will cause different drug pressures on the parasite genome. The presence of SP-resistant genotypes reflects the use of SP in the past and indicates a misguided use of SP for treating vivax malaria in those countries where SP is currently used. Nevertheless, the findings in our study are limited in scope because samples were collected over a 10-year period and malaria drug policies of each country may have changed during this period. The travel history associated with the clinical samples was not always complete and visits by these patients to multiple endemic countries at the time of infection cannot be excluded. In addition, patients’ clinical profiles were not available, and drugs used for treatment could not be determined. Sample numbers from various countries such as Thailand, Solomon Islands, South Korea and Cambodia were not adequately represented.

## 4. Materials and Methods 

### 4.1. Ethics Statement and Sample Collection

Clinical blood samples were collected during the time period of 2008–2018 from patients diagnosed with malaria at NSW Health Pathology-ICPMR, Westmead Hospital. At the time of diagnosis, all patients had recently been to malaria-endemic countries, such as India, PNG, Pakistan, Indonesia, Thailand, Solomon Islands, South Korea and Cambodia. There was one patient with no specific country of travel recorded because the patient had travelled to multiple countries within Southeast Asia (SEAN). *Plasmodium vivax* was diagnosed in these samples by an experienced microscopist and if the microscopic results were unclear, a nested Polymerase Chain Reaction (PCR) was used to determine the malaria species [61]. These clinical blood samples were used with the approval of the human ethics review committee for research in human subjects, research and development office, Western Sydney Local Health District (WSLHD) Research Governance Officer (HREC reference No. LNR/18/WMEAD/139 and SSA reference No. LNRSSA/18/WMEAD/140).

A total of 120 samples with *P. vivax* infections were included in this study and QIAamp mini DNA extraction kits were used to extract malaria DNA from the blood samples according to the manufacturer’s instructions. Extracted DNA was stored at −20 °C until required.

### 4.2. Sequencing of The Multidrug Resistance 1(Pvmdr1) Gene, The Putative Transporter Protein (Pvcrt-o) Gene, The Dihydrofolate Reductase (Pvdhfr) Gene and The Dihydropteroate Synthetase (Pvdhps) Gene

Specific oligonucleotide primers listed in Table 5 were used to amplify the *Pvmdr1* and *Pvcrt-o* genes by conventional PCR. The total volume of 50 µL for the PCR reaction mixture contained 0.2 mM of each dNTP, 1x of PCR buffer, 1 µM of each primer, 2.5 mM of MgCl_2_, 2 U of Taq polymerase and 3–5 µL of extracted sample DNA. Reaction cycle conditions were as follows: 95 °C for 5 min, then 35 cycles of [94 °C for 30 s, 59 °C for 45 s, 72 °C for 1.3 min] and the final step was 72 °C for 7 min. 

A nested polymerase chain reaction (nested PCR) was used to amplify the *Pvdhfr* and *Pvdhps* genes with two pairs of oligonucleotide primers shown in Table 5. The outer PCR reactions were performed in a total volume of 25 µL per reaction containing: 0.2 mM of each dNTP, 1x of PCR buffer, 0.2 µM of each primer, 1.75 mM of MgCl_2_ and 1 U of Taq polymerase. Extracted DNA (3 µL) was added to this PCR which was run under the following conditions: 94 °C for 5min, then 20 cycles of [94 °C for 30 s, 58 °C for 30 s, 68 °C for 60 s] and final annealing at 68 °C for 5 min. Products from the outer PCR reaction (2 µL) were added into the nested PCR reaction which contained 2 mM of dNTP mix, 1x of PCR buffer, 0.2 µM of each primer, 1.75 mM of MgCl_2_ and 1 U of Taq polymerase. The following reaction condition was used: 94 °C for 5 min followed by 40 cycles of [94 °C for 30 s, 58 °C for 30 s, 68 °C for 45 s] with a final step at 68 °C for 5 min. All amplicons were analysed by electrophoresis on a 1.5% agarose gel.

The amplicons of *Pvmdr1, Pvcrt-o, Pvdhfr* and *Pvdhps* were sequenced by Macrogen in South Korea. All DNA sequences were assembled using the Sequencher 5.3 program. The assembled DNA sequences were aligned and translated to amino acid sequences using the MEGA 7 program and compared to Genbank reference strains (Reference No. Pvdhfr: XM001615032 [62] and X98123.1 [27], Pvdhps: XM001617159 [62], Pvcrt-o: AF314649 [12] and Pvmdr: AY618622 [18]) in order to identify the presence of amino acid changes in the expected protein. All nonsynonymous mutations in the targeted genes identified in this study are shown in Table 6 and Table 7.

## 5. Conclusions

High prevalence of chloroquine-resistance-associated genotypes highlights the severe situation of chloroquine resistance in *P. vivax* in Oceania and South Asian and Southeast Asian regions. Genetic monitoring for-drug resistant *P. vivax* in travellers returning to Australia allows pre-empting rising treatment failures in these endemic countries. Furthermore, this data will serve as an early warning for imported cases of chloroquine-resistant strains of *P. vivax* so that alternative drug therapies can be instigated. The finding of mutations in *Pvdhfr* and *Pvdhps* genes suggests that SP drug pressure on *P. vivax* is already present, even though this drug is not recommended for use against *P. vivax* in these regions.

## Figures and Tables

**Table 1 pathogens-09-00101-t001:** Demographics of patients included in this study.

Gender	Age							Total Number
	≥10	11–20	21–30	31–40	41–50	51–60	61≤	
Male	1	9	27	17	7	12	15	88
Female	2	0	7	1	1	5	7	23
Unknown	0	0	2	0	1	1	5	9

**Table 2 pathogens-09-00101-t002:** Prevalence of target mutations in the multidrug resistance 1 (*Pvmdr1*) gene and the putative transporter protein (*Pvcrt-o*) gene, amongst return travellers to NSW, Australia during 2008–2018.

Genotype	Mutations	Regions			Total (N = 120)
		SA	SEA	Oceania	
		n = 73	n = 10	n = 37	
*Pvmdr1*					
	**Wildtype**	9 (12%)	0	0	9 (7.5%)
	Y976:F1076				
	Single Mutation	7 (9.5%)	0	0	7 (5.8%)
	**F**976				
	Single Mutation	50 (68%)	5 (50%)	8 (21.6%)	63 (52.5%)
	**L**1076				
	Double Mutations	2 (2.5%)	5 (50%)	23 (62%)	30 (25%)
	**F**976:**L**1076				
*Pvcrt-o*					
	**Wildtype**	49 (67%)	6 (60%)	22 (59%)	77 (64%)
	**K**10 **Insertion**	9 (12%)	2 (20%)	0	11 (9%)

South Asia (SA): India and Pakistan; Southeast Asia (SEA): Indonesia, Thailand, Cambodia plus South Korea (includes one case with exposure to multiple countries of SEA (SEAN)); Oceania: Papua New Guinea and Solomon Islands.

**Table 3 pathogens-09-00101-t003:** Prevalence of SP resistant-associated genotypes in the dihydrofolate reductase (*Pvdhfr*) gene and the dihydropteroate synthetase (*Pvdhps*) gene, amongst return travellers to NSW, Australia during 2008–2018.

Genotype	Mutations	Regions			Total (N = 120)
		SA	SEA	Oceania	
		n = 73	n = 10	n = 37	
*Pvdhfr*					
	**Wildtype**	43 (59%)	3 (30%)	7 (19%)	53 (44%)
	F57:S58:T61:S117:I173				
	Single Mutation	3 (4%)	1 (10%)	1 (10%)	5 (4%)
	**N**117				
	Double Mutations	18 (24.5%)	2 (20%)	1 (2.7%)	21 (17.5%)
	**R**58:**N**117				
	Double Mutations	1 (1%)	0	5 (13.5%)	6 (5%)
	**L**57:**R**58				
	Quadruple Mutations	2 (2.5%)	4 (40%)	21 (56.7%)	27 (22.5%)
	**L**/**I**57:**R**58:**M**61:**T**117				
*Pvdhps*					
	**Wildtype**	62 (84.9%)	5 (50%)	35 (94.5%)	102 (85%)
	S382:A383:K512:A553:V585				
	Single Mutation	2 (2.7%)	1 (10%)	0	3 (2.5%)
	**G**383				
	Single Mutation	1 (1%)	1 (10%)	0	2 (1.6%)
	**G**553				
	Double Mutations	6 (8%)	2 (20%)	1 (2.7%)	9 (7.5%)
	**G**383:**G**553				

South Asia (SA): India and Pakistan; Southeast Asia (SEA): Indonesia, Thailand, Cambodia plus South Korea (includes one case with exposure to multiple countries of SEA (SEAN)); Oceania: Papua New Guinea and Solomon Islands.

**Table 4 pathogens-09-00101-t004:** The prevalence of tandem repeat variants in the dihydrofolate reductase (*Pvdhfr*) gene.

Countries	*Pvdhfr gene*		
	Type 1	Type 2	Type 3
**SA** **(n = 73)**	55 (75%)	9 (12%)	4 (5%)
**SEA** **(n = 10)**	6 (60%)	3 (30%)	1 (10%)
**Oceania** **(n = 37)**	16 (43%)	-	19 (51%)
**Total** **(N = 120)**	77 (64%)	12 (10%)	24 (20%)

South Asia (SA): India and Pakistan; Southeast Asia (SEA): Indonesia, Thailand, Cambodia plus South Korea (includes one case with exposure to multiple countries of SEA (SEAN)); Oceania: Papua New Guinea and Solomon Islands.

**Table 5 pathogens-09-00101-t005:** Primers used to amplify the following *P. vivax* gene targets: the multidrug resistance 1 (*Pvmdr1*) gene, the putative transporter protein (*Pvcrt-o*) gene, the dihydrofolate reductase (*Pvdhfr*) gene and the dihydropteroate synthetase (*Pvdhps*) gene.

Name	Sequences 5’–3’	Gene Targets	Product Size	Tm. (°C)	References
Pvdhfr (outer) F Pvdhfr (outer) R	CACCGCACCAGTTGATTCCT CCTCGGCGTTGTTCTTCT	*Pvdhfr*	979	67.3 63.0	[25]
Pvdhfr (nested) F Pvdhfr (nested) R	CCCCACCACATAACGAAG CCCCACCTTGCTGTAAACC	*Pvdhfr*	755	61.5 64.2	[25]
Pvdhps (outer) F Pvdhps (outer) R	GATGGCGGTTTATTTGTCG GCTGATCTTTGTCTTGACG	*Pvdhps*	1009	62.8 58.5	[25]
Pvdhps (nested) F Pvdhps (nested) R	GCTGTGGAGAGGATGTTC CCGCTCATCAGTCTGCAC	*Pvdhps*	731	58.2 63.5	[25]
Pvcrt-o F Pvcrt-o R	CAGTGAGAAGCCCCTGTTCG CCGCTCATCAGTCTGCAC	*Pvcrt-o*	750	67.2 68.5	*
Pvmdr F Pvmdr R	GCGAACTCGAATAAGTACTCCCTCTA GGCGTAGCTTCC CGTAAATAAA	*Pvmdr1*	762	65.4 64.8	[4]

* A new primer set was designed by the authors to detect mutations in the resistant marker. (Tm **=** annealing temperature).

**Table 6 pathogens-09-00101-t006:** Nonsynonymous mutations associated with chloroquine resistance in a multidrug resistance 1 (*Pvmdr1*) gene and a putative transporter protein (*Pvcrt-o*) gene.

Genes	Nucleotide Position	Nucleotide Change	Amino Acid Position	Amino Acid Change
*Pvmdr1*	2928–2930	TAC -> T**T**C	976	(Y) Tyrosine -> (**F**) Phenylalanine
		(g.2929A>T)		(Y976**F**)
	3228–3230	TTT -> **C**TT	1076	(F) Phenylalanine -> (**L**) Leucine
		(g.3228T>C)		(F1076**L**)
*Pvcrt-o*	30	**AAG**	10	(**K**) Lysine (Insertion)
		(g.30_31insAAG)		(**K**10 insertion)

*Pvmdr1* = the multidrug resistance 1 gene; *Pvcrt-o* = the putative transporter protein gene.

**Table 7 pathogens-09-00101-t007:** Nonsynonymous mutations associated with sulphadoxine resistance in dihydrofolate reductase (*Pvdhfr*) gene and nonsynonymous mutations associated with pyrimethamine resistance in dihydropteroate synthetase (*Pvdhps*) gene.

Genes	Nucleotide Position	Nucleotide Change	Amino acid Position	Amino Acid Change
*Pvdhfr*	171–173	TTC -> TT**G**, TT**A**, **C**TC	57	(F) Phenylalanine -> (**L**) Leucine
		(g.173C>G, g.173C>G, g.173C>A, g.171T>C)		(F57**L**)
	171–173	TTC -> **A**T**A**	57	(F) Phenylalanine -> (**I**) Isoleucine
		(g.171T>A and 173C>A)		(F57**B**)
	172–174	AGC -> AG**G**, AG**A**	58	(S) Serine -> (**R**) Arginine
		(g.174C>G, g.174C>G)		(S58**R**)
	183–185	ACG ->A**T**G	61	(T) Threonine -> (**M**) Methionine
		(g.184C>T)		(T61**M**)
	351–353	AGC -> A**C**C	117	(S) Serine -> (**T**) Threonine
		(g.352G>C)		(S117**M**)
	351–353	AGC -> A**A**C	117	(S) Serine -> (**N**) Asparagine
		(g.352G>A)		(S117**N**)
*Pvdhps*	1149–1151	GCC -> G**G**C	383	(A) Alanine -> (**G**) Glycine
		(g.1150C>G)		(A383**G**)
	1659–1661	GCC -> G**G**C	553	(A) Alanine -> (**G**) Glycine
		(g.1660C>G)		(A553**G**)

*Pvdhfr* = the dihydrofolate reductase gene; *Pvdhps* = the dihydropteroate synthetase gene.

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
