# Peer review of "Molecular Detection of Antimalarial Drug Resistance in Plasmodium vivax from Returned Travellers to NSW, Australia during 2008–2018"

_pathogens, 2020, doi:10.3390/pathogens9020101_

Round 1

Reviewer 1 Report

Noisang et al. in this study investigated polymorphisms in four P. vivax genes from clinical isolates. These isolates were collected from travellers who visited P.vivax-endemic regions of Asia or Oceania and returned to Australia where they received hospital treatment.

Using this sample set, the authors investigated previously described SNPs associated with resistance to chloroquine and sulphadoxine-pyrimethamine (SP) treatment. This study is, as the authors describe themselves, based on a small sample size of P.vivax- infected travellers with travel histories to Asia and Oceania only, but nevertheless highlights important surveillance possibilities of emerging drug resistance in those P. vivax populations. With this information more effective treatment regimens can be implemented and for that reason this study is important and contributes to the increasing number of reports highlighting the spread of chloroquine treatment failure in P. vivax-endemic regions.

I would suggest to the authors to consider the following points:

To describe the SNPs in the four drug resistance-associated genes analysed here the authors used a very atypical and hard to follow nomenclature. Significant efforts have been made to standardize the nomenclature of genomic modifications and I would urge the authors to follow the guidelines described here (http://varnomen.hgvs.org/recommendations/DNA/).

Therefore p.[(Tyr976Phe; Phe1076Leu)] would be the correct way to describe the mdr1 double mutant.

the authors do not mention treatment outcome in the NSW hospital here, assuming that none of the travellers either received chloroquine based anti-malaria treatment or that no information was available. This is important to mention here. I would additionally urge the authors to turn down their language, in the abstract section especially. In line 32, the manuscript says that SP is “misused” in the treatment of vivax infections in the region, which is misleading, as it suggests that all 8 countries have implemented SP treatment to treat P. vivax malaria cases in the clinic, rather than SP treatment being used as a consequence of misdiagnosis (Pf instead of Pv) or because of co-infections with P. falciparum. In line 146 the authors state a report from 1989 as the first report of vivax resistance to chloroquine, which was based on a study in PNG. Is that true as previous reports in the literature exist from India (Talib, 1979) and Colombia (Arias and Corredor; 1989)? Lines 169-170 the authors write: “the occurrence of the double F976:L1076 mutations is a strong indicator…” Especially in light of the recent Sa et al, Nature Communications paper which shows that wild type sequence crt-o appears linked to increased chloroquine tolerance in vivax and that in their genetic cross a locus containing crt-o is the only resistance associated genomic locus, this should be toned down and the Sa et al paper should be cited in the discussion of this manuscript.

Author Response

REVIEWER 1:

Noisang et al. in this study investigated polymorphisms in four P. vivax genes from clinical isolates. These isolates were collected from travellers who visited P.vivax-endemic regions of Asia or Oceania and returned to Australia where they received hospital treatment.

Using this sample set, the authors investigated previously described SNPs associated with resistance to chloroquine and sulphadoxine-pyrimethamine (SP) treatment. This study is, as the authors describe themselves, based on a small sample size of P.vivax- infected travellers with travel histories to Asia and Oceania only, but nevertheless highlights important surveillance possibilities of emerging drug resistance in those P. vivax populations. With this information more effective treatment regimens can be implemented and for that reason this study is important and contributes to the increasing number of reports highlighting the spread of chloroquine treatment failure in P. vivax-endemic regions.

I would suggest to the authors to consider the following points:

To describe the SNPs in the four drug resistance-associated genes analysed here the authors used a very atypical and hard to follow nomenclature. Significant efforts have been made to standardize the nomenclature of genomic modifications and I would urge the authors to follow the guidelines described here (http://varnomen.hgvs.org/recommendations/DNA/).

Therefore p.[(Tyr976Phe; Phe1076Leu)] would be the correct way to describe the mdr1 double mutant.

Response:

Table 6 have been revised to include the accepted nomenclature as described by http://varnomen.hgvs.org/recommendations/DNA/. The description of nomenclature in buckets such as (Y976F)*, (F1076L)* , and (F57L)* were followed previous studies (Anantabotla et al., 2019; R Suwanarusk et al., 2008; Rossarin Suwanarusk et al., 2007; Tantiamornkul, Pumpaibool, Piriyapongsa, Culleton, & Lek-Uthai, 2018).

The authors do not mention treatment outcome in the NSW hospital here, assuming that none of the travellers either received chloroquine based anti-malaria treatment or that no information was available. This is important to mention here.

Response:

The information on each patient was not sufficient to provide further detail on the clinical treatment and subsequent outcome of each patient. There has not been any evidence of treatment failure in any of Vivax malaria cases examined in this study. However, information on the drugs used for each patient was not available.  We have added this limitation to lines 293-294 of our Discussion.

I would additionally urge the authors to turn down their language, in the abstract section especially. In line 32, the manuscript says that SP is “misused” in the treatment of vivax infections in the region, which is misleading, as it suggests that all 8 countries have implemented SP treatment to treat vivax malaria cases in the clinic, rather than SP treatment being used as a consequence of misdiagnosis (Pf instead of Pv) or because of co-infections with P. falciparum.

Response:

We accept the comments of the reviewer and have changed Line 32 to “turn down” our misleading language.

In line 146 the authors state a report from 1989 as the first report of vivax resistance to chloroquine, which was based on a study in PNG. Is that true as previous reports in the literature exist from India (Talib, 1979) and Colombia (Arias and Corredor; 1989)? Lines 169-170 the authors write: “the occurrence of the double F976:L1076 mutations is a strong indicator…” Especially in light of the recent Sa et al, Nature Communications paper which shows that wild type sequence crt-o appears linked to increased chloroquine tolerance in vivax and that in their genetic cross a locus containing crt-o is the only resistance associated genomic locus, this should be toned down and the Sa et al paper should be cited in the discussion of this manuscript.

Response:

The emergence of drug resistant Vivax malaria occurred in different countries over several years. We cannot provide a precise date to when Vivax malaria developed resistance and have decided to broaden the period when several publications reported Vivax resistance. Therefore, line 61-62 have been edited.

 We have also included the reference papers as recommended by Reviewer 1 and discussed the Sá, J.M., et al. paper in line 233-235.

Reviewer 2 Report

pathogens-635936: Molecular detection of antimalarial drug resistance in Plasmodium vivax from returned travellers to NSW, Australia during 2008-2018

The authors studied drug resistance in Plasmodium vivax by looking at mutations in Pvmdr1, Pvcrt-o, Pvdhfr, and Pvdhps genes in 120 blood samples from patients who had recently been to one of eight malaria-endemic countries and diagnosed with P. vivax infection. Overall the work is interesting despite its simplicity. However, a number of issues should be addressed before this work can be found publishable. 

Major Issues 

[1] The authors did not explain how and why do these mutations result in the Chloroquine resistance. There are mutations, then what? How do these mutations prevent chloroquine from having the antimalarial effect? 

[2] The authors stated that the drug-resistance mutations in the Plasmodium vivax (Pv) are analogous to those in Plasmodium falciparum (Pf). How do they compare? Can the authors show the results in a tabular form to compare the mutations leading to drug resistance in Pv and in Pf?

[3] The authors make the case that this work is important and can help for early warning. However, the authors were only able to get the results by using data from over a period that spans 10+ years. How is it feasible that this approach can aid early warning of drug-resistant when it is based on data collected over a very long period of time. This needs to be addressed. 

Minor Issues 

[4] Some sentences do not make much logical sense. For example, in the abstract, the authors wrote: "The continued use of chloroquine in these countries will likely lead to an increase in clinical cases of chloroquine resistance in P. vivax." How is this correct most especially that this work has looked at genes? Are you suggesting that the chloroquine itself is causing the mutations?  

A similar issue is found with this statement "The presence of the mutations associated with sulphadoxine-pyrimethamine resistance indicates that sulphadoxine-pyrimethamine is being misused in the management of P. vivax infections in these regions"

[5] There are issues with some sentence structures. For example "malaria endemic countries" should be "malaria-endemic countries", "found among of the samples" should be "found among the samples", "for detection of resistance" should be "for the detection of resistance", etc. These and all other grammatical issues should be corrected. 

Author Response

Major Issues 

The authors did not explain how and why do these mutations result in the Chloroquine resistance. There are mutations, then what? How do these mutations prevent chloroquine from having the antimalarial effect? 

Response:

The explanation has been added in Introduction between line 65 and 81 on page 2. The mutant genes associated with chloroquine resistance in P. vivax are orthologous to the mutant genes in P. falciparum (Rossarin Suwanarusk et al., 2007), which encodes a transmembrane protein located on the parasite digestive vacuole (Nomura et al., 2001; Parija & Praharaj, 2011). The protein is involved in the survival of the parasite at therapeutic levels of chloroquine.

The authors stated that the drug-resistance mutations in the Plasmodium vivax (Pv) are analogous to those in Plasmodium falciparum (Pf). How do they compare? Can the authors show the results in a tabular form to compare the mutations leading to drug resistance in Pv and in Pf?

Response:

Another supplementary table is added to the manuscript comparing resistance markers of P. falciparum and P. vivax (please see the supplementary table 4)

The authors make the case that this work is important and can help for early warning. However, the authors were only able to get the results by using data from over a period that spans 10+ years. How is it feasible that this approach can aid early warning of drug-resistant when it is based on data collected over a very long period of time. This needs to be addressed. 

Response:

While the work presented is only a collection of clinical samples obtained from travellers, it does allow us to examine retrospectively whether or not there is evidence of progression to increased resistance expressed in the parasite gene targets. This is the suggestion of the two-step mutation hypothesis (Please see lines 199-205). Although statistical analysis (fisher's exact test) was applied to 10 years of collected data, the low sample numbers spread across several countries did not show significance. We hope that this information can be used as data contributing to the global malaria surveillance program.

Minor Issues 

Some sentences do not make much logical sense. For example, in the abstract, the authors wrote: "The continued use of chloroquine in these countries will likely lead to an increase in clinical cases of chloroquine resistance in P. vivax." How is this correct most especially that this work has looked at genes? Are you suggesting that the chloroquine itself is causing the mutations?

A similar issue is found with this statement "The presence of the mutations associated with sulphadoxine-pyrimethamine resistance indicates that sulphadoxine-pyrimethamine is being misused in the management of P. vivax infections in these regions"

Response:

Lines 30-41 have been re-written to improve clarity.

There are issues with some sentence structures. For example "malaria endemic countries" should be "malaria-endemic countries", "found among of the samples" should be "found among the samples", "for detection of resistance" should be "for the detection of resistance", etc. These and all other grammatical issues should be corrected. 

Response:

These grammatical issues at line 23, 28 and 317 have been corrected.

Reviewer 3 Report

Molecular Detection of Antimalarial Drug Resistance in Plasmodium Vivax from Returned Travellers to NSW, Australia during 2008–2018. pathogens-635936.

The authors presented the molecular data of P. vivax, importance given the risk of resistance spreading. The literature survey of the study is not well conducted as the study carried out over a period of 10 year. The followings are the major concern of the MS.

It would be worth mentioning about the national drug policy in the countries where the samples were used since 2008. As it seems majority of the samples were from only three countries ie India, PNG and Pakistan so at-least mentioned for these countries. Description of molecular markers of anti-malarial which is not associated with clinical correlation is making deviations from the main theme of the study. Author may not have to access full clinical profile of the patients but whatever the information is available needs to be discuss otherwise it may be mentioned in the study limitation. Author should give the progressive trends of the resistance markers as the study was used samples over a 10 year. I would expect more from author in terms of reversal in the mutation and the role of resistance. It would be better if the data analysed based on the all four markers to find out correlation the National drug policy. Although the raw data is given in the supplementary table, it could be analysed by year as well as by mutations pattern. It would be better if the author come up with possible conclusive mechanism of arising and spreading of resistance with scientific evidence. I am a bit confused after reading the MS as it is kind of very good report but not a conclusive straightforward message and there is no statistical tests were used anywhere.

Author Response

Reviewer 3:

The authors presented the molecular data of P. vivax, importance given the risk of resistance spreading. The literature survey of the study is not well conducted as the study carried out over a period of 10 year. The followings are the major concern of the MS.

It would be worth mentioning about the national drug policy in the countries where the samples were used since 2008. As it seems majority of the samples were from only three countries ie India, PNG and Pakistan so at-least mentioned for these countries.

Response:

We accept that there are different policies for each country and should align our discussion accordingly. The drug policies for the most frequent number of cases (PNG, India and Pakistan) were added to the manuscript at line 204-206 on page 6 and 217-219 on page 7.

Description of molecular markers of anti-malarial which is not associated with clinical correlation is making deviations from the main theme of the study. Author may not have to access full clinical profile of the patients but whatever the information is available needs to be discuss otherwise it may be mentioned in the study limitation.

Response:

 The information on each patient was not sufficient to provide further detail on the clinical treatment and subsequent outcome. We have added this limitation to Line 293-295 of our Discussion.

Author should give the progressive trends of the resistance markers as the study was used samples over a 10 year.

Response:

We tried to apply a statistical analysis to the data, but unfortunately, our data was spread too widely between countries to provide any significant changes.

I would expect more from author in terms of reversal in the mutation and the role of resistance. It would be better if the data analyzed based on the all four markers to find out correlation the National drug policy.

Response:

We did not see a reversal in the mutation. Although chloroquine has not been used for 10 years in Papua New Guinea, we found more mutations associated with chloroquine resistance in P vivax.

Although the raw data is given in the supplementary table, it could be analysed by year as well as by mutations pattern. It would be better if the author come up with possible conclusive mechanism of arising and spreading of resistance with scientific evidence. I am a bit confused after reading the MS as it is kind of very good report but not a conclusive straightforward message and there is no statistical tests were used anywhere.

Response:

Supplementary Table 2 has been resorted according to the year of sample collection. We also consulted with a statistical epidemiologist. It was recommended that the fisher's exact test be used as a method for statistical analysis of this data. Unfortunately, when we tried to make a table of 3X3 there were zero numbers in the table (see below), this resulted in non-significant values over 0.05. Therefore, we chose to present our findings in a descriptive manner.

The 3x3 table of mutations associated with chloroquine resistance in P. vivax

SA

SEA

Oceania

Wildtype

9

0

0

Single Mutation

59

5

8

Double Mutation

2

5

23

Round 2

Reviewer 2 Report

The authors have addressed the issues raised. 

Author Response

Response: Spelling, typographical and grammatical errors have corrected

Reviewer 3 Report

The author have address all the queries.

Author Response

Response: those minor issue has been solved.